# Transdermal Delivery of Macromolecules Using Two-in-One Nanocomposite Device for Skin Electroporation

**DOI:** 10.3390/pharmaceutics13111805

**Published:** 2021-10-28

**Authors:** Juliette Simon, Bastien Jouanmiqueou, Marie-Pierre Rols, Emmanuel Flahaut, Muriel Golzio

**Affiliations:** 1Institut de Pharmacologie et de Biologie Structurale, IPBS, Université de Toulouse, CNRS, UPS, Université Toulouse 3—Paul Sabatier, 205 Route de Narbonne, CEDEX 4, 31077 Toulouse, France; juliette.simon@univ-tlse3.fr (J.S.); bastien.jouanmiqueou@ipbs.fr (B.J.); rols@ipbs.fr (M.-P.R.); 2Centre Interuniversitaire de Recherche et d’Ingénierie des Matériaux, CIRIMAT, Université de Toulouse, CNRS, Université Toulouse 3—Paul Sabatier, 118 Route de Narbonne, CEDEX 9, 31062 Toulouse, France

**Keywords:** skin electroporation, macromolecule delivery, carbon nanotubes, hydrogel composite

## Abstract

Delivery of hydrophilic molecules through the skin using electroporation is a promising alternative approach to intradermal injection. Recently, we developed a two-in-one electrode/reservoir material composed of carbon nanotubes and agarose hydrogel. In this work, we evaluated the potential of the device to achieve non-invasive transdermal drug delivery using skin electroporation. As it involved an electrode configuration different from the literature, critical questions were raised. First, we demonstrated the efficiency of the device to permeabilize the skin of hairless mice, as observed by propidium iodide (PI) uptake in the nuclei of the epidermis cells through macro fluorescence imaging and histology. Application of Lucifer yellow (LY) at different times after unipolar electroporation treatment demonstrated the partial reversibility of the skin permeabilization after 30 min, and as such, that barrier function properties tended to be restored. We uncovered, for the first time to our knowledge, an intrinsic asymmetry of permeation pathways generated in the *stratum corneum* during treatment. Electrophoresis was here the main driving force for macromolecule delivery, but it competed with passive diffusion through the generated aqueous pathways for smaller molecules. Finally, we validated 4 kDa dextran labelled with fluorescein isothiocyanate (FD4) as a model molecule to optimize the electrical parameters, needed to improve macromolecule delivery.

## 1. Introduction

Transdermal delivery is a very interesting route to explore for drug delivery. Compared to other routes, it can be fast and allow longer-term delivery. As an example, it can be considered a suitable substitute for oral delivery when drug degradation occurring during digestion hinders its efficiency too much. However, to achieve transdermal delivery it needs to cross the *stratum corneum* (SC), the most external layer of the epidermis. The SC is composed of corneocytes and lipids organized in multilamellar bilayers, and plays a major role in the skin barrier properties. Consequently, skin is impermeable to the passive diffusion of most hydrophilic and high molecular weight drugs, leading to the use of needles for direct injection [1]. The crossing of that hydrophobic protective layer is the main challenge for the development of alternative transdermal delivery methods and particularly for those intended to be non-invasive. Through the years, many technologies were considered for transdermal delivery, relying on chemical enhancers, physical approaches such as ultrasound and iontophoresis, or microporation techniques involving microneedles, which were widely studied for the variety of composition and shaping they offer [2,3,4].

Among the physical methods, electroporation (EP) consists of the application of high voltage pulses resulting in the reversible destabilization of the cell bilayer membrane, allowing molecule exchanges. Its efficiency was demonstrated in vivo to potentiate the uptake of anticancer drugs or even DNA [5,6,7,8]. The application of EP to a complex structure like skin has been investigated since the last two decades, allowing us to have now a better insight of permeabilization mechanisms. According to Pliquett et al. [9], application of high voltage pulses induced stochastic pores formation within the SC multilamellar lipid bilayers. The formation of those pores, or aqueous pathways, combined with Joules effects, resulted in the expansion of permeable regions, called local transport regions (LTRs), whose size and number depended on pulses voltage and duration [9,10,11]. Several works also focused on the development of mathematical and numerical models [12,13,14]. The feasibility of skin EP for transdermal delivery was also widely studied. Lombry et al. [15] demonstrated that the transdermal delivery of molecules sometimes up to 40 kDa using EP was possible in vitro. Further investigations pointed out the importance of the electrical parameters’ optimization, as well as the physicochemical properties of the drug [10]. In vivo studies demonstrated that the delivery of fentanyl in rats was reduced from hours to a few minutes using skin fold configuration [16]. Enhanced delivery of fluorescein isothiocyanate-labelled 4 kDa dextran compared to passive diffusion was achieved using a patch, removed during EP with multiarray electrodes, and reapplied afterward [17]. However, for more convenience, electrode configuration and reservoir availability need to be optimized.

The use of carbon nanotubes-based nanocomposites for biomedical applications has recently been spreading. Carbon nanotubes (CNT) alone were shown to be promising for biosensors [18,19], drug delivery [20] and tissue engineering [21,22], but toxicity concerns prevent most of their utilization in clinical trials [23]. An alternative to make the best of CNT outstanding mechanical and electrical properties while avoiding CNT exposure to tissue is to embed them in a biocompatible matrix, such as hydrogels. These CNT-based nanocomposites exhibited an even wider application range, thanks to the 3D porous structure provided by the hydrogel matrix [24]. We drew inspiration from this strategy to address the non-invasive transdermal delivery challenge. We developed a two-in-one material, a CNT-agarose nanocomposite, which would play both roles of being electrodes for skin EP and a reservoir for the drug. The application of two of those nanocomposite electrodes, directly on skin, enabled us to perform simultaneous skin permeabilization and drug delivery. The utilization of such platforms with unipolar pulses allowed us to obtain promising preliminary ex vivo results in a previous work [25]. The electrical characterization of the nanocomposite was also presented elsewhere [26].

In this study, we went a step further in the validation of the developed nanocomposite as a suitable solution to achieve transdermal delivery of hydrophilic molecules and macromolecules through skin permeabilization. We validated the actual permeabilization of the skin using propidium iodide (PI) and evaluated using the Lucifer yellow (LY) transitory nature of the permeabilization, vital to achieve fully non-invasive delivery. We also investigated the effect of the size and the charge of the loaded molecules using fluorescein isothiocyanate (FITC)-labelled 4 kDa dextran (FD4) and charged derivatives as models for hydrophilic macromolecules. As a conclusion, we proposed a mechanistic explanation of the phenomena involved and further developments that would be needed to improve further this two-in-one platform for electro-stimulated delivery.

## 2. Materials and Methods

### 2.1. Platform Preparation

The nanocomposite synthesis was already reported elsewhere [25,26]. Briefly, a solution of agarose at 25 g/L was mixed at 90 °C with a suspension at 0.25 g/L in water of doubled-walled CNT synthesized in our lab [27]. Carboxymethylcellulose with ultra-low viscosity was used to ensure homogeneity of highly hydrophobic carbon nanotubes in the hydrogels [28] and to facilitate the dispersion steps of sonication and shear mixing. The suspension was casted into 10 mm-diameter cylindrical molds (2 mm height) and left to cool down at room temperature. The obtained hydrogels were fully dried at 30 °C and finally swelled into electroporation buffer containing fluorescent markers. Their final diameter was d = 8.4 ± 0.4 mm. A schematic representation of the process, adapted from [25], can be found Appendix A. Propidium iodide (PI), at 100 µM, was used here as a permeabilization marker. We used 4 kDa FITC Dextran (FD4), at 1 mM to mimic macromolecule size. Lucifer yellow (LY), at 1 mM, following the work of Leclerc et al. [29], was used to study the reversibility of the skin permeation and the influence of the molecule size. FD4 grafted with carboxymethyl groups to be negatively charged (FD4-CM) at 1 mM, and FD4 grafted with diethylaminoethyl groups to be positively charged (FD4-DEAE) at 1 mM, were used to study the influence of the molecule charge. Finally, we challenged our model molecules by comparing it to insulin-FITC (I-FITC) at 0.111 mM. We took into account the FITC substitution ratio in each molecule to ensure to have the same FITC molar concentration in the different solutions. All reagents and fluorescent markers were purchased from Sigma-Aldrich (St. Louis, MO, USA) and diluted in 10 mM phosphate buffer.

### 2.2. Electroporation Treatment on Mice Skins

Explanted skins from different mice strains were were obtained less than 1 h before experiment, depending on availability: Female nude Balb/c AnNRj-Foxn1nu mice (Janvier Labs, Le Genest-Saint-Isle, France) aged of 10 to 12 weeks and weighing between 20 and 24 g; female nude NMRI-nu mice (Janvier Labs, Le Genest-Saint-Isle, France) aged of 8 to 16 weeks and weighing between 25 and 35 g; female hairless SKH1 mice (Charles River, Écully, France) aged of 8 to 16 weeks and weighing between 25 and 35 g. Dorsal skins were placed on a gauze soaked with commercial PBS solution without Ca+ and Mg+ (Eurobio, Les Ulis, France) in a petri dish, SC facing upward. Molecule-loaded platforms were placed directly on top of the skins. We used a custom-made electroporation support to ensure a constant 0.5 cm gap between the edges of both platforms. Stainless steel cylinders were used to ensure a homogeneous contact between the platform and the skin, and to electrically connect the platforms to the generator. Unipolar square wave pulses were delivered by a ELECTRO cell B10 HVLV, (Betatech, Saint-Orens-de-Gameville, France): set voltage = 300 V, duration = 20 ms, frequency = 1 Hz, 8 pulses, inspired from [30]. In all our experiments, the cathode side was defined as the negative pole. Treated skins were rinsed three times with PBS before imaging to remove non-absorbed markers on the skin.

### 2.3. Mice Skin Surface Fluorescence Imaging

Samples were observed with an upright Macrofluo fluorescence microscope, EL6000 source (Leica Microsystems, Rueil-Malmaison, France), equipped with a Cool Snap HQ Camera (Roper Scientific, Ottobrunn, Germany), and Metamorph (Molecular Devices, Sunnyvale, CA, USA) image acquisition software was used. Samples were imaged by fluorescence using appropriate filters: Leica L5 filter (excitation filter BP: 480/40 nm, dich. 505LP, emission filter BP: 527/30 nm) was used for FITC and LY, and Leica 49,008 ET mCH/TR filter (excitation filter BP: 560/40 nm, dich. 585LP, emission filter BP: 630/75 nm) was used for PI imaging (exposure time = 1 s). All filters were purchased from Leica Microsystems, Rueil-Malmaison, France. Images were processed for contrast and brightness, and analyzed with ImageJ (National Institute of Health, Bethesda, MD, USA). Quantification of fluorescent marker uptake was corrected using a ratio between the mean intensity of treated and non-treated areas in the same image, to avoid any bias induced by skins auto-fluorescence variability. All following graphics display the calculated relative fluorescence intensity (RFI) of each marker.

### 2.4. Histology

Immediately after imaging, the treated areas were removed and fixed overnight at 4 °C in periodate-lysine-paraformaldehyde buffer (PLP). The PLP was rinsed three times with PBS and samples were left for 24 h in successive sucrose baths of 15% and 30%, before being embedded in OCT for cryo-conservation (Shandon Cryomatrix, Richard-Allan Scientific, Subsidiary of Thermo Fisher Scientific, Kalamazoo, MI, USA). Slices of 10-µm were sectioned and DAPI was used for nuclei labelling (Mounting Medium with DAPI—Aqueous, Fluoroshield, Abcam, Cambridge, UK). To evaluate the penetration depths of the different fluorescent markers, the slices were observed with an AxioImager M2 equipped with a plan apochromat 40×/0.95 korr M27 objective and an AxioCam 503 color camera (Zeiss, Jena, Germany). Slides were imaged by fluorescence using appropriate filters: Zeiss 96 HE BP (excitation filter BP: 370–410 nm, emission filter BP: 430–470 nm) was used for DAPI observation, Zeiss 38 HE Green Fluorescent Prot. (excitation filter BP: 450–490 nm, emission filter BP: 500–550 nm) was used for FITC and LY, and Zeiss 20 HE Rhodamine (excitation filter BP: 540–552 nm, emission filter BP: 575–640 nm) was for used PI. All filters were purchased from Zeiss, Jena, Germany.

### 2.5. Statistical Analysis

Data analyses were performed using GraphPad Prism 5.04 program (GraphPad Software, Inc., La Jolla, CA, USA) and data were expressed as means ± SEM for the number of experiments indicated in the legends of the figures. Multiple comparisons were performed using one-way analysis of variance (ANOVA) followed by post-tests: Turkey’s multiple comparisons or Dunnett’s comparison against control.

## 3. Results

### 3.1. Two-in-One Device Allows Skin Electroporation

In our previous work, we demonstrated that the proposed device was able to increase the uptake of FD4 in the skin [25]. In this work, we tried to deepen our understanding of the phenomena involved, and the first step was to demonstrate that we achieve skin permeabilization. To do so, platforms loaded with PI were used to treat hairless mice skins. The marking pattern evidenced differences, we observed an asymmetry of PI delivery between the anode and the cathode, but also a non-uniformity of the delivery into the skin below the electrodes (Figure 1a). The quantification of the relative fluorescence intensity (RFI) confirmed the observed increased uptake of PI on the anode side, compared to the natural uptake that was obtained after 30 min (Figure 1b). The uptake on the cathode side was not significantly different from the control (non-pulsed platform). We performed histological observations on those samples (Figure 1c). The presence of red nuclei in the epidermis on the anode side demonstrated that PI was able to cross the SC. On the other hand, red nuclei were rarely observed on the cathode side.

PI is a red DNA intercalant, which labels the nucleus of cells with a permeable plasma membrane. As such, it is not only a useful marker to detect and quantify electropermeabilization, but it can also be used to reveal cell death. To determine if the electroporation treatment was inducing cellular death, skins were treated with platform loaded with buffer only, and 30 min after, to allow potential resealing to occur, 100 µL of PI at 100 µM was applied. In parallel, the same amount of PI was directly applied on non-pulsed skin to compare the uptake. In both cases, PI was left for 30 min at 37 °C and rinsed before imaging. The quantification of PI RFI revealed no significant differences between the skin PI natural uptake and the uptake 30 min after treatment (Appendix A). We concluded that cell death was not involved in a significant way during our experiments. From this point, histological observations indicated that the device used with unipolar pulses allowed us to achieve skin permeabilization. However, an asymmetry of the delivery was still present and might be related to several phenomena. It could be the result of electrophoretic forces, pushing or pulling positively charged PI depending on the electrode. Our second hypothesis was that the use of unipolar pulses might generate asymmetric permeation structures in the skin, leading to more or less available pathways for the drug to be delivered.

### 3.2. Evaluation of the Skin Permeabilization Transitory Nature

The reversibility of the skin permeabilization is a compulsory feature to develop a fully non-invasive delivery method through EP. To evaluate it, we used a second fluorescent smaller molecule, Lucifer yellow (LY), already used in the literature to assess the state of skin barrier function [29]. Here, platforms loaded with electroporation buffer were used to perform EP treatment, and 100 µL of LY were spread on top of the treated area after 0, 15, 30 or 60 min. The samples were left for 1 h at 37 °C, before rinsing and imaging. The marking pattern showed an equivalent and intense marking at both electrodes when LY was applied directly after treatment (Figure 2a). The fluorescence intensity decreased at the anode when 15 min was waited before application, an equivalent pattern was observed for a 30 min and 60 min delay. RFI quantification was done for the different time points (Figure 2b). The results showed a decrease of the LY uptake at both electrodes as a function of time, which hinted that the skin tended to retrieve its original impermeability. Nevertheless, the return at the cathode side was not complete, as the uptake stagnated between 30 and 60 min after EP. On the contrary, the statistical analysis demonstrated that the skin original impermeability was reached after 15 min at the anode side (Figure 2b). This experiment also evidenced an asymmetric uptake between both electrodes. It was approximately 1.5 times higher at the cathode directly after treatment, and there was no more improved delivery at the anode side after 15 min. The LY uptake in this experiment occurred without the presence of an electric field, meaning the only driving force allowing fluorescent dye to penetrate the skin was passive diffusion. This result could imply that unipolar pulses generate different pathways depending on the electrode polarity.

### 3.3. Influence of the Molecule Size on Delivery

In order to observe the influence of the molecule size during skin electroporation-assisted delivery, we repeated the experiment using LY inside the platforms and compared with platforms loaded with FD4, which is ten times bigger than LY. We evaluated their marking pattern and we performed histological observations to evaluate their penetration depths into the skin. The delivery pattern demonstrated that both molecules were preferentially marking the cathode side on the contrary of PI (Figure 3a). This can be explained when considering the charges of the molecules; LY has a negative charge and the FITC group substituted on the dextran also introduces negative charge at neutral pH. Histological observations showed an important natural accumulation of LY in the SC (Figure 3b). The observation on the cathode side demonstrated that FD4 tended to accumulate in the SC at the cathode side, while LY could be found in the dermis upon treatment. Both molecules were not delivered similarly into the layers of the skin. Pathways exist, as demonstrated by the delivery of LY in the dermis. However, they might not yet be suitable for the delivery of molecules of bigger size, like FD4.

### 3.4. Influence of the Molecule Charge on Delivery

Our experiments brought into focus an inherent structural asymmetry between the two electrodes. However, questions about the influence of the electrophoretic forces on the loaded molecules during EP remained. To answer those questions, we compared our model molecule, FD4, with two of its derivate forms: one was a 4 kDa dextran grafted with carboxymethyl group to bring in negative charges (FD4-CM), and the other was grafted with diethylaminoethyl groups to positively charge it (FD4-DEAE). The FITC substitution ratio was the same for the three molecules and is two orders of magnitude smaller than the nitrogen or carboxymethyl substitution ratio. Therefore, FD4 charge was negligible when compared to its derivatives, and FD4 could be consider as a neutral molecule. The fluorescence macrographs of the differently charged FD4 showed different marking patterns (Figure 4a). Quantification of FITC RFI on treated skin revealed a doubled uptake of FD4-CM at the cathode and an equal uptake at the anode compared to FD4. We could also observe a weaker uptake of those two molecules at the anode side compared to the FD4 uptake on non-treated skin. An opposite behavior was observed for FD4-DEAE. Namely, the uptake was 30% weaker than FD4 at the cathode but seemed to be increased at the anode (Figure 4b). As a control experiment, the comparison of the uptake of each molecule on non-pulsed skin did not show statistically significant differences (Appendix A). Those behaviors were in agreement with the electrode’s polarity. In our configuration, the cathode was the negative and thus, electrophoretic forces dragged negatively charged molecules into the skin. On the contrary, they could be retained inside the platform at the anode side. We performed complementary experiments where the polarity of the electrodes was reversed between each pulse (Appendix A). No statistically significant difference of the labelling between the two electrodes was observed for FD4 nor FD4-CM molecules. The electrophoretic forces alternatively pushing and pulling the molecules depending on its current polarity could explain these results. Histological observations gave evidence of an accumulation of molecules in the SC, consistent in intensity and uniformity with the labelling already observed by macro-fluorescence imaging of the skin (Figure 4c).

### 3.5. Two-in-One Device Allowed Immediate Delivery with Skin Electroporation

To evaluate the contribution of electrophoresis and passive diffusion in the delivery process, the duration of contact between skin and platforms after treatment was increased. Platforms loaded with LY or FD4 were used for electroporation treatment and left in place for different durations, 0, 5, 15, 30 or 60 min, before rinsing and imaging. Results showed that the uptake at the cathode of pulsed skin for any duration was significantly superior to non-pulsed skins uptake (Figure 5). However, there was no evolution of the uptake of both molecules with time. Considering the anode side, the delivery of LY was not significantly modified by the EP treatment. A different behavior was observed for FD4 where the uptake at the anode was significantly decreased. Therefore, we could conclude that for a positively charged molecule, the delivery at the cathode side was immediate and mainly provided by electrophoresis.

### 3.6. Validation of FD4 as a Model Molecule for Insulin Delivery Using Skin Electroporation

FD4 was used as a model drug to mimic macromolecules, such as insulin. Nevertheless, dextran and insulin, as a glucose derivative and protein, have different chemical functions and a different global charge. In order to check if the simple approximation of the size was a sufficient criterion for FD4 to mimic the behavior of insulin during electroporation treatment, platforms were loaded with insulin labelled with FITC (I-FITC) and compared to FD4 and its negative derivative FD4-CM, in terms of delivery pattern and uptake on skin after EP treatment. I-FITC concentration was calculated to be the same than the FITC molar concentration of FD4 and FD4-CM. The three macromolecules had the same delivery pattern (Figure 6a), but we observed a significant uptake difference at the cathode between FD4-CM and I-FITC (Figure 6b). The mean fluorescence intensity of I-FITC was more similar to the mean fluorescence of the FD4 than the FD4-CM. Histological observation demonstrated that, like FD4, I-FITC tended to accumulate in the SC without crossing it (Figure 6c). Those results indicated that FD4 can be used for our future works on skin EP electrical parameters optimization.

## 4. Discussion

Transdermal delivery is a crucial route to explore in order to elaborate efficient treatment protocol; however, the development of non-invasive methods alternative to injection are challenged by the impermeable and hydrophobic *stratum corneum* (SC) skin layer. In this work, we evaluated the ex vivo application of a custom two-in-one nanocomposite material designed to be simultaneously a drug reservoir and an electrode for skin EP. The first experiment presented demonstrated the actual permeabilization of viable cells of the epidermis as their nuclei were labelled with PI (Figure 1b). Therefore, the electric field was sufficient in this area to destabilize the cell plasma membrane and allow PI uptake in the permeabilized cells, and SC was also permeable enough to allow PI to reach deeper layers. Additionally, we observed an asymmetry of PI uptake, that was hardly observed in cathode histology slices. We suggested two possible explanations for this asymmetry as we dismissed cellular death: the occurrence of structural asymmetry, or the effects of electrophoretic forces.

As skin barrier function assayed using LY was shown to partially return to normal, we considered the device could be non-invasive for on shot uses (Figure 2). This recovery was however incomplete, which was in accordance with the work of Dujardin et al. on skin EP [31]. They measured the impedance of rats’ skin in vivo after treatment with 10 pulses of 335 V—5 ms and they observed a dramatic decrease compared to the control. The skin impedance tended to increase with time after treatment, but the skin original state was still not recovered after six hours. However, a good recovery was observed using 10 pulses of 1000 V—100 µs. Similar results were obtained in vitro [32]. It would be necessary in further works to move to in vivo studies to evaluate more precisely the skin resealing after treatment. Indeed, ex vivo, the lack of blood circulation to provide nutriments to the tissue essential to maintain homeostasis could impair the recovery. Cycles of EP treatment, spaced by several hours, could be used to assess if skin EP induces irreversible damages in the SC and what would be the effects of its repeated utilization on the same area. Finally, the non-uniformity of the electrical field received by the skin, evidenced by the non-uniform delivery pattern in every macrograph presented in this work, could also suggest that the skin barrier properties were not affected or impaired equally over all of the 50 mm^2^ treated area. Impedance measurements performed with a smaller platform could help us to determine the voltage and pulses duration range to apply to stay in a fully reversible state. It would also be interesting to proceed to the optimization of those electrical parameters using in situ skin impedance measurements to get real-time feedback and immediately adjust parameters, as recently reported in the work of Atkins et al. [33].

We evidenced an uptake asymmetry between the two electrodes, which was independent from electrophoretic forces (Figure 2). Considering the work of Weaver et al. [11], the SC structure could be described with a brick-and-mortar model, where corneocytes were separated one from another by lipid bilayers. LTRs would form in those layers upon electroporation treatment, allowing hydrophilic molecules to go through the SC. To the best of our knowledge, we were the first to use this electrode configuration, and we suggested here that LTRs were not uniformly generated depending on the electrode polarity. They could be wider and/or more numerous at the cathode side, offering more available aqueous pathways for the drug to cross the SC. A schematic description is proposed in Figure 7. We are willing to perform in situ electrical measurements of the system device including the skin during EP treatment to investigate that theory. Those data will be useful in particular to build numerical model of our system.

Experiments demonstrated a clear influence of the molecule size and charge on the delivery, which was expected according to literature [10]. The permeabilization was sufficient to allow the delivery of LY in the dermis. These encouraging results allow us to move towards models closer to human skin for this range of molecule sizes (Figure 3). FD4 tended to accumulate in the SC, but that accumulation was nevertheless significantly improved upon EP treatment. The size of the molecule could be the limiting factor for now, ensued from the asymmetric SC permeabilization that has changed the balance between the two delivery forces, namely passive diffusion and electrophoresis. We observed that most of the delivery occurred during the application of the EP treatment; therefore, electrophoresis would be the major driving force (Figure 5), which is consistent with the work of Regnier et al. [34]. However, the utilization of loaded platforms for EP treatment showed different behaviors at the anode side between LY and FD4. LY uptake was unchanged upon treatment, while FD4 delivery was decreased. Considering the LY case, both delivery forces might have been nullifying each other. Indeed, results presented in Figure 2 demonstrated that when LY was applied after EP, its uptake was significantly increased at the anode. As such, LY delivery is possible through passive diffusion into the generated LTRs so opposing electrophoresis might result in an unchanged uptake. Considering FD4, its delivery was probably impaired by its size and blocked by SC at the anode side. Therefore, the electrophoresis was the only remaining force dragging the FD4 up in the platform and reducing the uptake below the negative control. This could be verified remembering the delivery of positively charged FD4-DEAE to the anode side presented Figure 4. We observed an increased delivery of FD4-CM compared to FD4 at the cathode, resulting from the addition of electrophoresis forces to passive diffusion, repulsing the negatively charged molecules from the negative electrode into the skin. We would expect the same phenomenon to occur on the anode side for positively charged molecules if the SC permeabilization was uniform between electrodes. However, the uptake of FD4-DEAE was not significantly different from the control; consequently, we concluded that smaller available permeable pathways at the anode side limited the delivery of the macromolecule. A schematic sketch of those phenomena depending on the molecules size and charge is presented in Figure 8.

To conclude, the device we presented in this work showed promising results of transitory skin permeabilization and immediate small molecule delivery into the dermis. We demonstrated that skin EP with unipolar pulses and non-invasive electrodes induced the asymmetrical generation of aqueous pathways (LTRs) in the SC between the electrodes side. Macromolecules, using FD4 as a model, could not yet be delivered across SC. However, the electrical parameters we used in this work were inspired from our previous work on intradermal gene delivery. As such, it exists a wide margin for optimization, aiming to improve SC permeabilization and macromolecules penetration depth by modifying pulses voltage and duration. The electrophoretic drag appeared to be a powerful lever to improve delivery, and it could be interesting to increase both pulse voltage and duration to intensify it. However, according to the work reported by Zorec et al. [35], short pulses of high voltage might be responsible for the generation of small defects with a high density. Meanwhile, lower voltage and longer pulses result in fewer, but wider LTR, which seems to be more efficient for delivery. Therefore, we might investigate the combination of different voltages and pulse duration, to first optimize LTRs generation and then push macromolecules thanks to electrophoresis, while preserving tissue integrity. Finally, the implementation of in situ electrical measurements to help us optimize electrical parameters might also help us to deepen our understanding of the delivery mechanisms, especially if combined with numerical models.

## Figures and Tables

**Figure 1 pharmaceutics-13-01805-f001:**
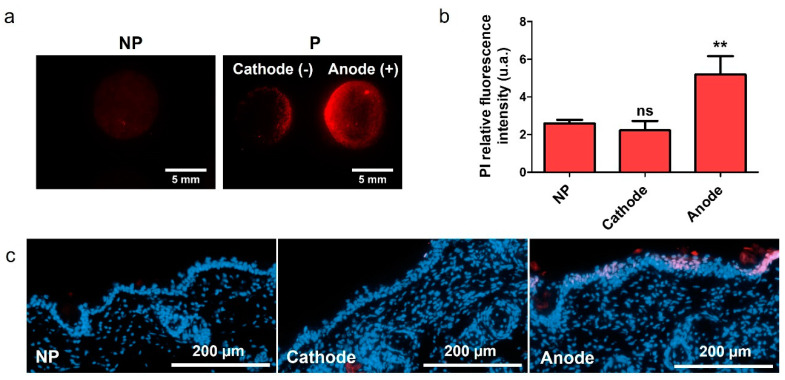
Evaluation of skin permeabilization using the two-in-one nanocomposite device for EP. Platforms loaded with PI (100 µM) were used to apply unipolar EP treatment on fresh hairless mice skin explants. Skins were imaged 30 min after. (**a**) Marking pattern of skins treated with unipolar-pulsed electric field, platforms loaded with PI. (**b**) Quantification of PI RFI at the cathode and anode of treated skins (P), compared to the PI RFI of non-pulsed (NP) skins. Statistical analysis: one-way ANOVA, with Dunnett’s post-test vs. NP. (Codes signification: ** = *p* > 0.01; ns = non-significant). *n* = 9. (**c**) Histological observations of 10 µm slices of skin treated with EP using platforms loaded with PI. PI is in red, and nuclei are labelled in blue by DAPI.

**Figure 2 pharmaceutics-13-01805-f002:**
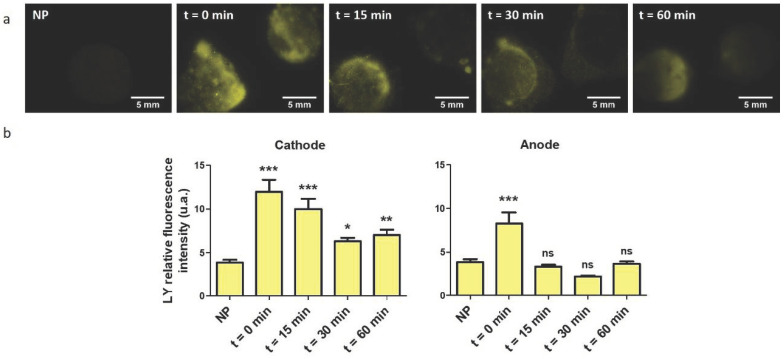
Evaluation of the skin permeabilization reversibility using the two-in-one nanocomposite device for EP. Platforms loaded with buffer only were used to apply unipolar EP treatment on hairless mice skin explants. One hundred microliters of LY (1 mM) were applied 0, 5, 15, 30 and 60 min after EP treatment, and left for 1 h at 37 °C to assess the evolution of skin permeability through time. (**a**) Delivery pattern of LY applied at 0, 5, 15, 30 and 60 min after treatment with unipolar-pulsed electric field, compared to non-pulsed (NP) skin, platforms loaded with buffer. Cathode side was always on the left. (**b**) Quantification of LY RFI at the cathode and anode side of treated skin, depending on the time waited before LY application, compared to the LY RFI of non-pulsed skin. Statistical analysis: one-way ANOVA, Dunnett post-test vs. NP data. (Codes signification: * = *p* > 0.05; ** = *p* > 0.01; *** = *p* > 0.001; ns = non-significant). Error bars indicate SEM. *n* = 3–4.

**Figure 3 pharmaceutics-13-01805-f003:**
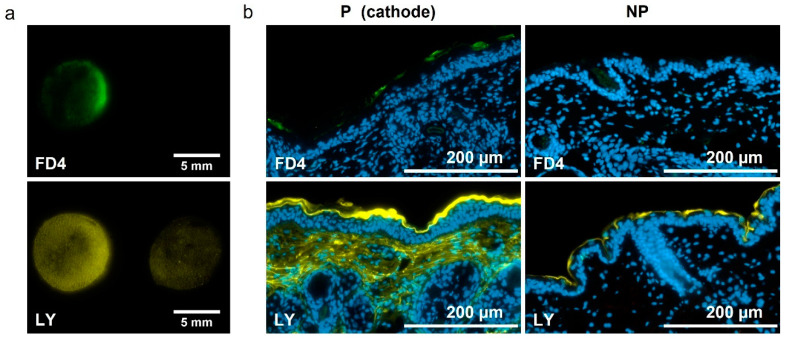
Influence of the molecule size on delivery. Platforms loaded with either LY or FD4 (1 mM), were used to apply unipolar EP treatment on hairless mice skin explants. Skins were imaged 30 min after. (**a**) Marking pattern of skins treated with unipolar-pulsed electric field, platforms loaded with either FD4 or LY. Cathode side is always on the left. (**b**) Histological observations of 10 µm slices of skins pulsed (P) with EP treatment or non-pulsed (NP), using platforms loaded with either FD4 or LY. FD4 is in green, LY in yellow, and nuclei are labelled in blue by DAPI.

**Figure 4 pharmaceutics-13-01805-f004:**
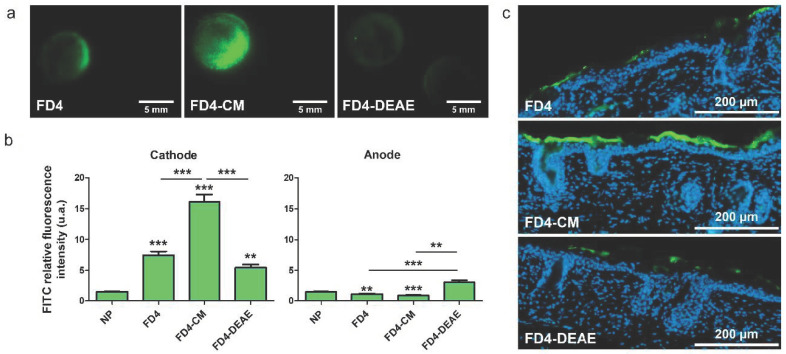
Influence of the molecule charge on delivery. Platforms loaded with either FD4, FD4 substituted with carboxymethyl groups (FD4-CM), or diethylaminoethyl groups (FD4-DEAE) (1 mM) were used to apply unipolar EP treatment on hairless mice skin explants. Skins were imaged 30 min after. (**a**) Marking patterns on skin of FD4, FD4-CM and FD-DEAE loaded on the platforms after treatment with a unipolar-pulsed electric field. Cathode is always on the left. (**b**) Quantification of FITC RFI at the cathode and anode side of treated skins, compared to the matched RFI on non-pulsed skins (NP). Statistical analysis: one-way ANOVA, Turkey’s post-test comparison between NP, FD4, FD4-CM and FD4-DEAE. (Codes signification: ** = *p* > 0.01; *** = *p* > 0.001; else is non-significant). Error bars indicate SEM. *n* = 4–9. (**c**) Histological observations of 10 µm slices of skin treated with EP using platforms loaded with either FD4, FD4-CM or FD4-DEAE. Only the cathodes are displayed. FITC is green and nuclei are labelled in blue by DAPI.

**Figure 5 pharmaceutics-13-01805-f005:**
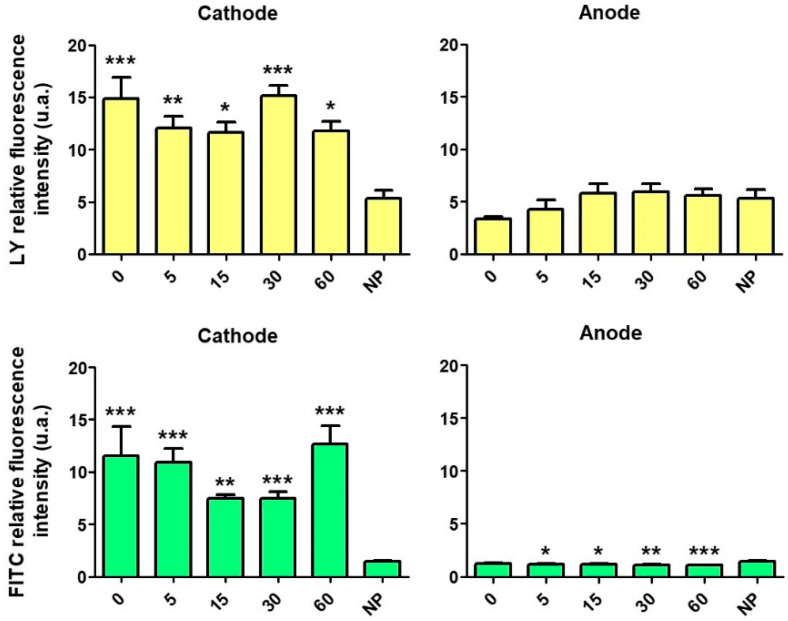
Effect of the duration of contact between skin and platforms after EP treatment. Platforms loaded with either LY or FD4 (1 mM), were used to apply unipolar EP treatment on hairless mice skin explants. The duration of contact between skin and platforms after treatment was increased to see the contribution of electrophoresis and passive diffusion. Quantification of LY and FITC RFI at the cathode and anode of skins treated with unipolar-pulsed electric field and platforms loaded with either LY or FD4. Platforms were left 0, 5, 15, 30 or 60 min on the skin after treatment. Statistical analysis: one-way ANOVA, Dunnett’s post-test vs. NP data. Codes signification: * = *p* > 0.05; ** = *p* > 0.01; *** = *p* > 0.001; else is non-significant. Error bars indicate SEM. *n* = 3–9.

**Figure 6 pharmaceutics-13-01805-f006:**
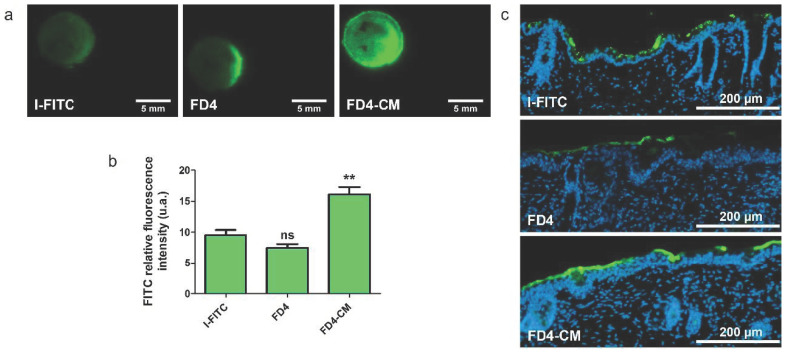
Comparison of FITC labelled insulin (I-FITC) with FD4 and FD4-CM behavior during delivery through skin EP. Platforms loaded with either I-FITC, FD4 or FD4-CM (1 mM), were used to apply unipolar EP treatment on hairless mice skin explants. Skins were imaged 30 min after. (**a**) Delivery pattern of I-FITC, FD4 and FD4-CM loaded in platforms after treatment with unipolar pulsed electric field. (**b**) Quantification of FITC RFI of the three molecules at the cathode of treated skin. Statistical analysis: one-way ANOVA, Dunnett post-test vs. I-FITC data. (Codes signification: ** = *p* > 0.01; ns = non-significant). *n* = 3–9. (**c**) Histological observations of 10 µm slices of skin treated with EP using platforms loaded with either FD4, FD4-CM or I-FITC. Only the cathodes are displayed. FITC is green and nuclei are labelled in blue by DAPI.

**Figure 7 pharmaceutics-13-01805-f007:**
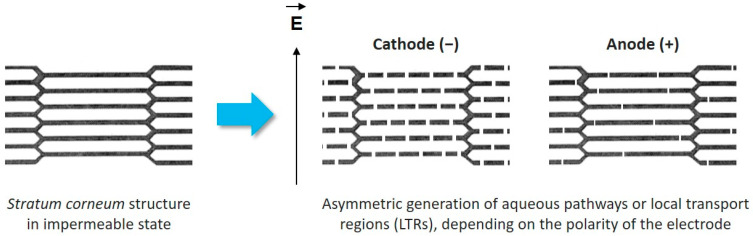
LTRs are generated differently depending on the electrode polarity upon application of EP treatment. They might be more numerous and/or wider at the cathode side. Adapted from [11].

**Figure 8 pharmaceutics-13-01805-f008:**
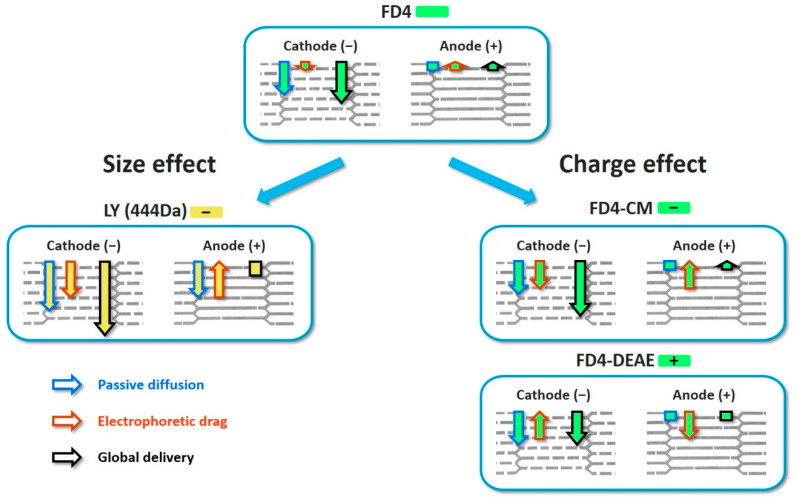
Scheme of the global delivery resulting from passive diffusion through aqueous pathways generated upon EP treatment and electrophoresis, depending on molecule size and charge. FD4 global charge was considered negligible compared to other presented molecules and results in a reduced electrophoresis contribution. At the cathode, LTRs were sufficient for FD4 to accumulate in the SC. Passive diffusion and electrophoresis combined for an improved delivery. At the anode, LTRs were not sufficient to allow passive diffusion, and the remaining electrophoresis opposed global delivery and reduced the uptake. LY is smaller than FD4 and the generated LTRs are sufficient for its passive diffusion at both electrodes. At the cathode, both forces combined to greatly improve the delivery while they nullified each other at the anode. For FD4-CM, the electrophoresis had a higher contribution, so the uptake was higher at the cathode, while unchanged at the anode, compared to FD4. Finally, FD4-DEAE charge was the opposite of the three other molecules. As such, the electrophoresis contribution on each side was reversed. It opposed passive diffusion at the cathode resulting in a decreased delivery compared to FD4. The delivery at the anode was however unaffected by the electrophoretic drag, since LTRs were not sufficient to allow passage to the macromolecule.

## Data Availability

Data are available upon request.

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
