# Peer review of "Transdermal Delivery of Macromolecules Using Two-in-One Nanocomposite Device for Skin Electroporation"

_pharmaceutics, 2021, doi:10.3390/pharmaceutics13111805_

Round 1

Reviewer 1 Report

The authors have shown systematic studies of electroporation-induced skin permeabilization with different molecules with varied size and charge. The experiments are well designed, and the data are quite clear to show the delivery efficacy. Therefore, I recommend this for publication. One question is what happens if you change the intensity of the electrical input, how would this further impact the penetration of the cargo?

Reviewer 2 Report

This is an interesting manuscript focused on the validation of permeabilization of the skin studies.  In general, the work is well written and clear. This topic is more than suitable for Pharmaceutics. However, several aspects should be checked before publication: 

  • Line 30-31 I agree that, at specific situation, transdermal delivery is a good alternative. However, it seems that its is for everything. I suggest changing or deleting this sentence.
  • Methods: Mice strain, weight, age...should be included.
  • Figure 1A and Figure 2A: Have both images the same intensity/brightness? If not, please adjust  (e.g using Image J)
  • I suggest including information about the developed nanoplatforms, at least as supplementary material
  • In my opinion the objective it is not so clear. I suggest rewriting that part
